# Dynamic Neural Regeneration: Enhancing Deep Learning Generalization on Small Datasets

**Vijaya Raghavan T Ramkumar[1,], Elahe Arani[1,2,], & Bahram Zonooz[1,*]**
[1]Eindhoven University of Technology    [2]Wayve
raghavijay95@gmail.com, e.arani@tue.nl, b.zonooz@tue.nl

## Abstract

The efficacy of deep learning techniques is contingent upon access to large volumes of data (labeled or unlabeled). However, in practical domains such as medical applications, data availability is often limited. This presents a significant challenge: How can we effectively train deep neural networks on relatively small datasets while improving generalization? Recent works have explored evolutionary or iterative training paradigms, which reinitialize a subset of parameters to enhance generalization performance for small datasets. However, these methods typically rely on randomly selected parameter subsets and maintain fixed masks throughout training, potentially leading to suboptimal outcomes. Inspired by neurogenesis in the brain, we propose a novel iterative training framework, Dynamic Neural Regeneration (DNR), that employs a data-aware dynamic masking scheme to eliminate redundant connections by estimating their significance. This approach increases the model's capacity for further learning through random weight reinitialization. Experimental results demonstrate that our approach outperforms existing methods in accuracy and robustness, highlighting its potential for real-world applications where data collection is challenging. [2]

## 1   Introduction

Deep neural networks (DNNs) have become essential for solving complex problems in various fields, such as image and speech recognition, natural language processing, and robotics (1). With the increasing availability of data, DNNs have achieved unprecedented performance, surpassing human-level performance in some applications (2). However, the success of DNNs is limited when dealing with small datasets, where the model tends to overfit and fails to generalize to new data. For example, it is often difficult to obtain a large amount of data in medical diagnosis due to the complexity and high cost of the procedures involved. In such cases, lack of generalization can be dangerous, which can lead to incorrect diagnosis and treatment.

Recently, several studies based on weight reinitialization methods (3; 4) have been proposed in the literature to improve generalization by iteratively refining the learned solution through partial weight reinitialization. These methods select and retain a subset of parameters while randomly reinitializing the rest of the network during iterative/evolutionary training schemes. For example, a state-of-the-art method named Knowledge Evolution (KE) (5) improves generalization by randomly splitting the network into fit and reset subnetworks and constantly reinitializing the reset subnetwork after each iteration. However, the KE approach is limited by its reliance on a predetermined mask creation, where a random subset of parameters is selected and kept constant throughout the iterative training process. This constraint may impede the model's ability to learn effectively from small datasets, ultimately limiting its generalization capabilities. These limitations raise two important questions:

---

*Equal contribution.
[2]Code is available at `https://github.com/NeurAI-Lab/Dynamic-Neural-Regeneration`

1) Can we leverage an evolutionary training paradigm to evolve or adapt the mask over generations, instead of using a fixed mask, in order to enhance the generalization performance of deep learning models trained on small datasets? 2) Can we utilize the available data and the internal state of the model to dynamically determine the important parameters for each generation, rather than randomly presetting them?

In our quest to address these questions, we draw inspiration from the phenomenon of neurogenesis in the brain. Neurogenesis, the process of dynamically generating or eliminating neurons in response to environmental demands, has been found to play a crucial role in learning and memory consolidation (6; 7; 8). This intricate process enables the brain to adapt to new experiences and stimuli, enhancing generalizability. Recent advances in neuroscience have shed light on the non-random integration of new neurons within the brain (9). For instance, in rodents, neurogenesis-dependent refinement of synaptic connections has been observed in the hippocampus, where the integration of new neurons leads to the elimination of less active synaptic connections (10; 11). Selective neurogenesis is critical to improving generalization ability by providing a diverse pool of neurons with distinct properties that can integrate into existing neural networks and contribute to adaptive learning (12). Although the precise mechanisms that govern selective neurogenesis are not fully understood, these findings suggest that selective neurogenesis in the human brain enhances generalization capabilities through its dynamic and selective nature. Thus, by emulating the characteristics of selective neurogenesis, we unlock its potential to improve generalization in deep neural networks.

Therefore, we present a novel iterative training approach called Dynamic Neural Regeneration (DNR), which distinguishes itself from the conventional Knowledge Evolution (KE) method through its mask computation. Unlike a predetermined fixed mask, DNR utilizes a data-aware dynamic masking criterion that evolves and adapts the mask over generations. Through extensive experiments on multiple datasets, we demonstrate that our proposed training paradigm greatly improves the performance and generalization of the models. Furthermore, DNR effectively addresses overfitting on relatively small datasets, alleviating the need for extensive data collection. The main contributions of the paper are as follows.

- Dynamic Neural Regeneration (DNR) is an evolutionary training paradigm that incorporates data-aware dynamic masking to selectively transfer knowledge across generations.
- Our proposed training paradigm facilitates the learning of generalizable features and increases the overall performance of DNNs across small datasets.
- DNR exhibits robustness in solving more common challenges in real-world problems, including class imbalance, natural corruption, and adversarial attacks.

## 2   Related work

Iterative training and weight reinitialization for DNNs is a prominent area of research (5; 4; 13; 14) that focuses mainly on improving generalization performance by partially refining the learned solution or fully iterating the learned solution. Dense-Sparse-Dense (DSD) (3) propose a three-phase approach where weights with small magnitudes are pruned after initial training to induce sparsity and retrain the network by reinitializing the pruned weights to zero. Zaidi et al. (15) conducted an extensive investigation into the conditions under which reinitialization proves beneficial. BANs (Born Again Neural Networks) (4) is a knowledge-distillation-based method that follows a similar iterative training paradigm. However, the critical difference between our work and BANs is that it employs the class-logits distribution instead of the network weights to transfer knowledge between successive networks. Recently, Zhou et al. (16) (LLF) propose the forget and relearn hypothesis, which aims to harmonize various existing iterative algorithms by framing them through the lens of forgetting. This approach operates on the premise that initial layers capture generalized features, while subsequent layers tend to memorize specific details. Accordingly, they advocate for the repeated reinitialization and retraining of later layers, effectively erasing information related to challenging instances. Similarly, the LW (17) approach progressively reinitializes all layers. However, these weight reinitialization methods rely on architecture-specific assumptions that do not take the data into account. They are based on presumed properties inherent to the model and its learning process. Consequently, these methods lack prior knowledge of which features, layers, or parameters should be reinitialized in a general context. Moreover, research indicates that memorization in neural networks is not confined to final layers but involves neurons distributed throughout the model (18). Knowledge Evolution (KE) (5) splits model

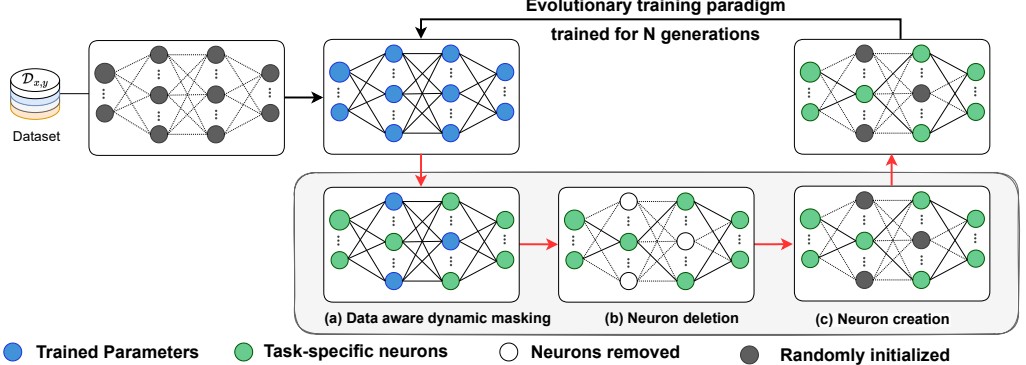

Figure 1: Schematics of proposed *Dynamic Neural Regeneration (DNR)* framework. Our framework utilizes a data-aware dynamic masking scheme to remove redundant connections and increase the network's capacity for further learning by incorporating random weight reinitialization. Thus, effectively improving the performance and generalization of deep neural networks on small datasets.

weights into fit and reset parts randomly and iteratively reinitializes the reset part during training. The splitting method can be arbitrary (weight-level splitting (WELS)) or structured (Kernel-level convolutional-aware splitting (KELS)). This approach involves perturbing the reset hypothesis to evolve the knowledge within the fit hypothesis over multiple generations. Our framework (DNR) distinguishes itself from the conventional Knowledge Evolution (KE) method through its mask computation. DNR utilizes data-aware dynamic masking that adapts the mask over generations and transfers selective knowledge.

Moreover, we differentiate our study from current literature on neural architecture search (NAS) (19) and growing neural networks (20). Our emphasis is on a consistent network architecture, maintaining fixed connections and parameter count throughout our analysis. Notably, our work sets itself apart from the dynamic sparse training literature (21; 22), as our objective is not to achieve sparsity but rather to enhance generalization on small datasets.

## 3 Methodology

### 3.1 Evolutionary training paradigm

We first introduce the evolutionary/iterative training paradigm as envisioned in KE (5). Evolutionary training paradigms allow neural networks to be trained for many generations, where each generation focuses on optimizing the model to converge towards a local minimum while progressively improving generalization. Each generation within the training process is denoted as $g$, where $g$ ranges from 1 to the total number of generations, $N$.

We define a deep neural network $f$ with $L$ layers and is characterized by the set of parameters $\Theta$. We assume a dataset $D$ consisting of $n$ input-output pairs, denoted $\{(x_i, y_i)\}_{i=1}^{n}$. For a classification task, we define the cross entropy loss for training the network as:

$$\mathcal{L}_{ce} = -\frac{1}{n} \sum_{i=1}^{n} [y_i \log(\text{softmax}(f(x_i; \Theta))) + (1 - y_i) \log(1 - \text{softmax}(f(x_i; \Theta)))] \quad (1)$$

where $\hat{y}_i = f(x_i)$ is the network's predicted output for input $x_i$. We initialize the weights and biases of the network randomly.

KE starts by introducing a binary mask $M$, which partitions the weights of the neural network into two hypotheses before starting training: the fit hypothesis $H_{\text{fit}}$ and the reset hypothesis $H_{\text{reset}}$. This partitioning is expressed as follows:

$$H_{\text{fit}} = \mathbf{M} \odot \Theta \quad \text{and} \quad H_{\text{reset}} = (\mathbf{1} - \mathbf{M}) \odot \Theta \quad (2)$$

Here, the element-wise multiplication operator $\odot$ is applied to the mask $M$ and the parameter set $\Theta$ to obtain the fit hypothesis $H_{\text{fit}}$. Similarly, the reset hypothesis $H_{\text{reset}}$ is obtained by element-wise

multiplying the complement of the mask $(1 - M)$ with the parameter set $\Theta$. These parameters are chosen at random before the start of the first generation. This binary mask M is kept constant throughout the evolutionary training; i.e., the parameters belonging to the fit and reset hypotheses remain in that category across all generations.

We use the stochastic gradient descent (SGD) algorithm to train the network with a learning rate $\alpha$. We run SGD for $e$ epochs on the dataset $D$. The beginning of every new generation is characterized by introducing perturbations applied to the network weights to induce a high loss. This is done by reinitializing the parameters in the reset hypothesis while transferring or retaining the parameters belonging to the fit hypothesis. This dynamic process triggers a subsequent round of optimization, guiding the neural network toward the search for a new minimum in the parameter space. The initialization of the network $f$ for the next generation $f_g$ is as follows:

$$\Theta_g \leftarrow \mathbf{M} \odot \Theta_{g-1} + (\mathbf{1} - \mathbf{M}) \odot \Theta_{Reinit} \tag{3}$$

where $\Theta_{g-1}$ and $\Theta_g$ are the parameters of the network $f$ belonging to the previous generation and current generation, respectively. $\Theta_{Reinit}$ corresponds to the randomly initialized tensor sampled from a uniform distribution. We then train the next generation of the network $f_g$ using SGD with the same hyperparameters and epochs as the first generation.

## 3.2 Dynamic Neural Regeneration (DNR) with Data-aware Dynamic Masking

Unlike KE, we propose a methodology that offers a distinct advantage regarding binary mask computation and parameter reinitialization. Motivated by the symbiotic link between generalization and selective neurogenesis in biological neural networks (9), we introduce a *Data-aware Dynamic Masking* (DDM) that emulates the process of selective neurogenesis in evolutionary training. The benefits of DDM's way of reinitialization are two-fold. 1) It takes advantage of the evolutionary training paradigm and adapts the mask dynamically in each generation rather than using a predetermined mask. This introduces flexibility in the network and improves the generalization performance of deep learning models trained on small datasets. 2) Our masking scheme leverages a model's data and internal state to dynamically determine the important parameters for a given task, rather than relying on random pre-setting, to enhance the performance of deep learning models on small datasets. Our way of masking offers a priori knowledge of where and what parameters and layers should be reinitialized in the general case.

The mask $M$ is calculated at the beginning of each generation in a data-dependent manner. We assess the importance or sensitivity of each connection in the network to the specific task by employing the SNIP method (23). SNIP decouples the connection weight from the loss function to identify relevant connections. We randomly sample a small subset of data ($\pi$) from the current dataset to evaluate connection sensitivity. We define a connection sensitivity mask $\mathbf{M} \in \{0, 1\}^{|\Theta|}$, where $|\Theta|$ denotes the number of parameters in the network. The mask is designed to maintain a sparsity constraint $k$, which specifies the percentage of parameters to retain. The computation of connection sensitivity is performed as follows:

$$\mathcal{G}_j(\mathbf{\Theta}; \pi) = \lim_{\delta \to 0} \frac{\mathcal{L}_{ce}(\mathbf{M} \odot \mathbf{\Theta}; \pi) - \mathcal{L}_{ce}\left((\mathbf{M} - \delta \mathbf{e}_j) \odot \mathbf{\Theta}; \pi\right)}{\delta} \Bigg|_{\mathbf{M}=1} \tag{4}$$

where $j$ corresponds to the parameter index and $e_j$ is the mask vector of the index $j$, where the magnitude of the derivatives is then used to calculate the saliency criteria ($s_j$):

$$s_j = \frac{|\mathcal{G}_j(\mathbf{\Theta}; \pi)|}{\sum_{k=1}^{m} |g_k(\mathbf{\Theta}; \pi)|}. \tag{5}$$

After calculating the saliency values, we apply the sparsity constraint $k$ to the connection sensitivity mask, which ensures that only the top-k task-specific connections are retained. The sparsity constraint $k$ is defined as follows:

$$\mathbf{M}_j = \mathbb{1}\left[s_j - \tilde{s}_\kappa \geq 0\right], \quad \forall j \in \{1 \ldots m\}, \tag{6}$$

where $\tilde{s}_k$ is the $k^{\text{th}}$ largest element in the saliency vector $s$, $\mathbb{1}[.]$ is the indicator function and m represents the total number of parameters in the neural network. Subsequently, using the saliency values obtained from the connection sensitivity analysis, we select and preserve the top-k important connections. The parameters associated with the connections deemed less important for the current

Table 1: Compares the results of our method with the other weight reinitialization methods on ResNet18. $g$ in $f_g$ indicates the number of generations the model is trained.

| Methods | Small Datasets | | | | | |
| --- | --- | --- | --- | --- | --- | --- |
| | CUB | Aircraft | Dog | Flower | MIT | Mean |
| CE ($f_1$) | $53.57_{\pm0.20}$ | $51.28_{\pm0.65}$ | $63.83_{\pm0.12}$ | $48.48_{\pm0.65}$ | $55.28_{\pm0.19}$ | 54.49 |
| DSD | $53.00_{\pm0.32}$ | $\mathbf{57.24}_{\pm0.21}$ | $63.58_{\pm0.14}$ | $51.39_{\pm0.19}$ | $53.21_{\pm0.37}$ | 55.68 |
| BAN ($f_{10}$) | $53.71_{\pm0.35}$ | $53.19_{\pm0.22}$ | $64.16_{\pm0.13}$ | $48.53_{\pm0.17}$ | $55.65_{\pm0.28}$ | 55.05 |
| KE ($f_{10}$) | $58.11_{\pm0.25}$ | $53.21_{\pm0.43}$ | $64.56_{\pm0.31}$ | $56.15_{\pm0.19}$ | $58.33_{\pm0.43}$ | 58.07 |
| DNR ($f_{10}$) | $\mathbf{59.72}_{\pm0.21}$ | $55.87_{\pm0.47}$ | $\mathbf{65.76}_{\pm0.13}$ | $\mathbf{58.10}_{\pm0.24}$ | $\mathbf{61.78}_{\pm0.36}$ | 60.25 |
| Smth ($f_1$) | $58.92_{\pm0.24}$ | $57.16_{\pm0.91}$ | $63.64_{\pm0.16}$ | $51.02_{\pm0.09}$ | $57.74_{\pm0.39}$ | 57.70 |
| Smth + LW ($f_8$) | $70.50_{\pm0.26}$ | $67.10_{\pm0.32}$ | $65.76_{\pm0.36}$ | $66.92_{\pm0.20}$ | $61.67_{\pm0.30}$ | 66.39 |
| Smth + LLF ($f_8$) | $\mathbf{71.30}_{\pm0.14}$ | $\mathbf{68.87}_{\pm0.12}$ | $66.35_{\pm0.22}$ | $67.20_{\pm0.24}$ | $63.14_{\pm0.18}$ | 67.37 |
| Smth + DNR ($f_8$) | $70.95_{\pm0.16}$ | $66.10_{\pm0.25}$ | $\mathbf{66.56}_{\pm0.18}$ | $\mathbf{68.50}_{\pm0.27}$ | $\mathbf{63.94}_{\pm0.21}$ | 67.21 |
| Smth + LB ($f_{10}$) | $69.80_{\pm0.13}$ | $65.29_{\pm0.51}$ | $66.19_{\pm0.03}$ | $66.89_{\pm0.23}$ | $61.29_{\pm0.49}$ | 65.89 |
| Smth + KE ($f_{10}$) | $66.51_{\pm0.070}$ | $63.32_{\pm0.30}$ | $63.86_{\pm0.21}$ | $62.56_{\pm0.17}$ | $59.58_{\pm0.62}$ | 63.17 |
| Smth + DNR ($f_{10}$) | $\mathbf{71.37}_{\pm0.22}$ | $\mathbf{66.63}_{\pm0.37}$ | $\mathbf{66.81}_{\pm0.20}$ | $\mathbf{68.36}_{\pm0.14}$ | $\mathbf{64.10}_{\pm0.58}$ | 67.45 |

generation are then reinitialized. This process effectively induces selective neurogenesis, allowing the network to adapt and free up its capacity for learning more generalized representations in subsequent generations. Finally, the network for subsequent generation is initialized as shown in Equation 3.

Intuitively, we incorporated selective neurogenesis as a replacement mechanism, reinitializing the input and output synaptic weights of specific subsets of network parameters dynamically during the evolutionary training process (24). Due to the challenges associated with where, how, and when to create neurons (20), we explore data-aware dynamic masking to drive neuron creation and removal, which could improve learning. We first select the crucial parameters based on the computed saliency mask. Ideally, we would like the mask to keep the knowledge learned from the previous generation as much as possible and to have enough learning capacity to accommodate the learning happening in the new generation. The additional learning capacity facilitates the fast adoption of generalized knowledge and reinforces the knowledge retained from the previous generation. In this way, selective neurogenesis is achieved that inherently adapts the network connectivity patterns in a data-dependent way to learn generalized representations without altering overall network size.

The network with the new initialization undergoes next-generation training with the same data for the $e$ epochs, where $e$ is kept the same for each generation. The network is trained with the loss function shown in Equation 1. Thus, we favor the preservation of the task-specific connections more precisely than the mask criteria used in KE that can guide the network towards those desirable traits that efficiently improve the performance and generalization of DNNs in small datasets.

## 4 Experimental Setup

Here, we provide the details on the experimental setup, implementation details, and datasets used in our empirical evaluation.

**Datasets:** We evaluate the proposed method using five datasets: Flower102 (25), CUB-200-2011 (26), MIT64 (27), Stanford Dogs (28), FGVC-Aircraft (29). The summaries of the statistics of the data set are mentioned in Appendix, Table 10.

**Implementation Details:** Since our framework is a direct extension of the KE, we follow the same experimental setup. The efficacy of our framework is demonstrated in two widely used architectures: ResNet18 and ResNet50 (30). We randomly initialize the networks and optimize them with stochastic gradient descent (SGD) with momentum 0.9 and weight decay $1e-4$. We use the cosine learning rate decay with an initial learning rate lr = {0.1, 0.256} on specific datasets. The networks are trained iteratively for $N$ generations ($N$=11) with a batch size $b$=32 for $e$=200 epochs without early stopping. The standard data augmentation techniques, such as flipping and random cropping, are used. We employ SNIP (23) with network sparsity $k$ to find the critical subset of parameters at the end of each

generation. For the importance estimation, we use 20% of the whole dataset as a subset ($\pi$). For all our experiments, we reinitialize a fixed 20% parameters of the network globally. All training settings (lr, $b$, $e$) are constant throughout generations.

**Baselines:** To evaluate and benchmark the effectiveness of our proposed approach, we conduct a comprehensive evaluation by comparing it against several existing methods that involve iterative retraining and reinitialization. Specifically, we benchmark our method against the following techniques: 1) Dense-Sparse-Dense Networks (DSD) (3); 2) Born Again Networks (BANs) (4); and 3) Knowledge Evolution (KE) with its variant, KELS (5). We also compare our method against a non-iterative approach known as the Long Baseline (LB), which undergoes training for the same number of epochs as the corresponding iterative methods. Since our framework is built on top of KE, we follow the same procedure in all our experiments unless specified.

## 5   Results

In this section, we conduct comprehensive experimental evaluations of our method across multiple datasets and ablation studies. For an in-depth analysis, extended robustness experiments, including robustness to 15 types of natural corruptions at varying severity levels, adversarial attacks, and class imbalance, are provided in Appendix.

### 5.1   Evaluation on Small Datasets

Table 1 presents the quantitative classification evaluation results using ResNet18. $f_g$ denotes the result at the end of $g^{th}$ generation. We compare DNR with two different configurations: (1) using naive cross-entropy loss (CE), and (2) incorporating label smoothing (Smth) regularizer with a hyperparameter $\alpha = 0.1$ (31).

The Dynamic Neural Regeneration (DNR) framework demonstrates flexibility and consistently improves performance over the considered baselines across datasets. Interestingly, KE underperforms in terms of performance compared to long baseline (LB) with equal computation cost. This discrepancy may be attributed to the use of fixed masking criteria throughout evolutionary training, limiting the model's adaptability. In contrast, DNR outperforms both longer baselines and KE, consistently improving generalization performance across all datasets.

Similarly, we compare the performance of our method with the label smoothing regularizer (Smth) applied to the baselines. Table 1 shows that our method consistently outperforms LB and KE on all datasets. DNR showcases slightly better performance compared to LW and LLF, with a key advantage being its independence from architecture-specific assumptions. Unlike LLF and LW, which lack prior knowledge of which features, layers, or parameters to reinitialize, DNR uses data-aware dynamic masking to selectively remove redundant connections. This reduces complexity and improves scalability as the model grows. Moreover, research indicates that memorization in neural networks is not confined to final layers but involves neurons distributed throughout the model (18), highlighting the limitations of architecture-specific assumptions and underscoring the effectiveness of our approach. Additionally, in real-world settings where data arrives in batches (32), DNR can dynamically iterate over each batch, optimizing performance. These results demonstrate the efficacy of the data-aware dynamic masking and selective reinitialization employed by DNR. By adapting task-specific parameters in each generation, DNR achieves superior performance and enhances the model's generalization.

### 5.2   Evaluation on Large Datasets

Our work is a direct extension of KE (5), which focuses explicitly on improving generalization in the low data regime. However, we also thoroughly evaluate our method on large datasets such as Tiny-ImageNet (33), CIFAR10, and CIFAR100 (34) using ResNet50 to assess its scalability. Table 2 compares the effectiveness of our method (DNR) with Knowledge Evolution (KE) and longer baseline (LB) in larger data sets. For each model, we trained it on top of the baseline for a specific number of generations ($f_{10}$), where N indicates the number of generations. The proposed approach exhibits promising performance and generalization across various large-scale datasets, such as TinyImageNet, when compared to KE and longer baselines. Furthermore, while the performance of KE and longer baselines (LB) falls below the normal standard training ($f_1$), the DNR framework demonstrates

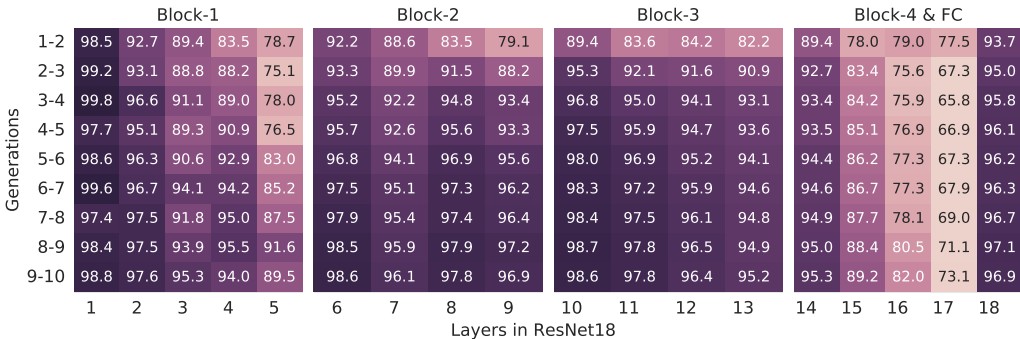

Figure 2: Layer-wise percentage overlap of the retained parameters in consecutive generations.

Table 2: Compares the results of the DNR framework with the KE and longer baselines for ResNet50 on large datasets. $g$ in $f_g$ indicates the number of generations the model is trained.

| Methods | Large datasets | | | |
|---|---|---|---|---|
| | CIFAR10 | CIFAR100 | TinyImageNet | Mean |
| Smth ($f_1$) | $94.32_{\pm0.38}$ | $73.83_{\pm0.23}$ | $54.15_{\pm0.18}$ | 74.10 |
| Smth + LB ($f_{10}$) | $93.60_{\pm0.29}$ | $74.21_{\pm0.28}$ | $51.16_{\pm0.21}$ | 72.99 |
| Smth + KE ($f_{10}$) | $93.50_{\pm0.22}$ | $73.92_{\pm0.31}$ | $52.56_{\pm0.17}$ | 73.33 |
| Smth + DNR ($f_{10}$) | $\mathbf{94.61}_{\pm0.30}$ | $\mathbf{75.05}_{\pm0.23}$ | $\mathbf{54.50}_{\pm0.26}$ | 74.72 |

comparable or slightly improved performance in this scenario. This suggests that a selective way of reinitializing benefits iterative training and can effectively handle the complexities and challenges associated with larger datasets and architectures.

## 5.3  Robustness of Connection Selection across Training Steps

Unlike KE, which employs a randomly predetermined and fixed masking strategy, DNR provides a notable advantage through the utilization of Data-aware Dynamic Masking (DDM) for parameter reinitialization. Therefore, it is crucial to investigate whether DNR fully leverages the benefits of the evolutionary training paradigm by dynamically adapting the mask in each generation.

The proposed DNR framework employs SNIP (23) as a masking criterion to selectively regulate the parameters that have the least impact on performance in each generation of training. To examine this, we analyze the CUB200 dataset using the ResNet18 architecture. We save the mask generated by SNIP after the end of every generation. Visualizing the mask generated by the DNR framework can be challenging due to the large number of parameters in each layer of the backbone. To assess the consistency of connections across generations, we adopt a metric based on the percentage of overlap of retained parameters between the masks created in consecutive generations. This metric provides a quantitative analysis of the degree of flexibility induced by DNR in the evolutionary training process.

Figure 2 illustrates the layer-wise percentage overlap of retained parameters between consecutive generations in the DNR framework. The results reveal that the earlier layers consistently exhibit a high overlap percentage across all generations, indicating a consistent selection of connections.

The overlap percentage decreases in the later layers (specifically, layer 4 in ResNet) as the model learns class-specific information. This observation suggests that the mask adapts to capture task-specific features while maintaining stability in the earlier layers. Interestingly, we observe that the overlap percentage of the mask progressively increases as the evolutionary training progresses. Specifically, the overlap between the 9th and 10th generations is higher compared to the overlap between the 1st and 2nd generations. This observation suggests that the mask becomes more saturated and stable as the model state converges to a lower-loss landscape. This flexible nature of the DNR framework, allowing for the regulation of connections in both early and later layers, contributes to its effectiveness in improving generalization performance.

Table 3: Comparative analysis of DNR and transfer learning across diverse datasets.

| Baselines | CUB | Aircraft | Dog | Flower | MIT |
|---|---|---|---|---|---|
| Smth + transfer learning (f3) | 65.63 ±0.21 | 61.02 ±0.23 | 63.84 ±0.17 | 57.62 ±0.19 | 58.04 ±0.31 |
| Smth + DNR (f3) | **68.56** ±0.24 | **64.37** ±0.19 | **65.72** ±0.15 | **62.13** ±0.23 | **62.62** ±0.51 |

## 5.4 Comparison with Transfer Learning

Our approach using DNR, indeed differs widely from the domain of transfer learning. Unlike transfer learning, which primarily focuses on leveraging pre-trained models trained on large datasets from different domains to boost task performance on downstream tasks, DNR is intricately designed to tackle the intricate challenge of enhancing generalization in the presence of inherently limited or small datasets. A key issue with transfer learning arises when the pre-trained model's source domain vastly differs from the target domain of interest. This discrepancy between domains often leads to domain shifts, where the knowledge transferred from the pre-trained model fails to adapt well to the specificities of the target domain, thereby resulting in suboptimal performance.

In particular, in scenarios like medical applications, obtaining sufficient labeled data that closely aligns with the task at hand is exceptionally challenging. Though transfer learning is predominantly used in this field, the need for domain expertise, privacy concerns, and the uniqueness of each application domain make it exceedingly difficult to find a pre-trained model that seamlessly fits. Furthermore, the presence of domain shift between the source and target might lead to compromised performance, affecting the accuracy and generalization of the model on a specific task with limited data.

DNR, on the other hand, offers a novel solution to these intricate challenges. By employing data-aware dynamic masking and selective reinitialization, DNR fosters the gradual evolution of the network, enabling it to adapt more effectively to the characteristics of the specific dataset. This process circumvents the problems of domain shifts that often plague transfer learning methods. Thus, while transfer learning remains valuable in contexts with abundant and well-aligned data, DNR stands out as a specialized approach to address the unique hurdles faced in scenarios of limited data availability, where the domain-shift problem can severely hinder model performance and generalization.

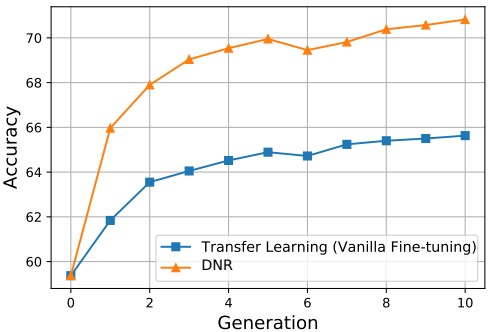

Figure 3: Convergence behavior: evaluating performance across generations in DNR and transfer learning with ResNet18 architecture trained on CUB dataset.

Furthermore, we have included a comparative analysis in Table 3 that involves an instance of transfer learning within the iterative training process, a process we refer to as vanilla fine-tuning. In this particular case, weights are directly transferred from one generation to the next without undergoing reinitialization.

This comparison serves to highlight the unique effectiveness of the DNR method. Our results distinctly demonstrate that DNR enhances the process of generalization, showcasing superior performance in comparison to the approach of directly transferring the complete network's weights across generations. This outcome further underscores the distinct advantage of DNR in evolving the network's capacity for better adaptation and learning in the evolutionary training paradigm.

## 5.5 Analyzing Convergence Patterns: A Comparative Study between DNR and Transfer Learning

In Figure 3, we present the convergence behavior of our proposed DNR algorithm juxtaposed with vanilla fine-tuning. The x-axis delineates different generations during the training process, while the y-axis represents the performance at the end of each generation. We observe that the convergence of vanilla fine-tuning unfolds at a more gradual pace. In contrast, DNR demonstrates a faster

convergence rate. Across generations, DNR consistently surpasses vanilla fine-tuning, delivering a heightened performance level within a shorter training duration. The utilization of data-aware dynamic masking through SNIP in DNR amplifies this efficiency, enabling the model to concentrate on the most pertinent information for effective generalization.

### 5.6 Effect of Importance Estimation Method

We conduct an investigation into the effectiveness of different methods to estimate the importance of parameters within our proposed training paradigm. Specifically, we explore Fisher Importance (FIM), weight magnitude, random selection, and SNIP (23) criteria. In Table 4, we present the performance and generalization results of the model trained with these various selection methods on the CUB200 dataset using the ResNet18 architecture.

Table 4: Evaluating the performance of DNR with different importance estimation.

| Importance Criteria | CUB200 | Flower |
|---|---|---|
| LB | $69.80_{\pm 0.13}$ | $66.89_{\pm 0.23}$ |
| Random (KE) | $66.51_{\pm 0.07}$ | $62.56_{\pm 0.17}$ |
| FIM | $67.73_{\pm 0.28}$ | $65.96_{\pm 0.20}$ |
| Weight Magnitude | $64.18_{\pm 0.19}$ | $66.90_{\pm 0.11}$ |
| SNIP | $\mathbf{71.87}_{\pm 0.22}$ | $\mathbf{68.36}_{\pm 0.14}$ |

Our findings demonstrate that the use of SNIP as data-aware dynamic masking yields superior performance compared to all other baseline methods. Surprisingly, the importance criteria based on weight magnitude exhibited inferior performance compared to random selection. However, the lottery ticket hypothesis (35) suggests the existence of sparse subnets within neural networks. Remarkably, when these subnets are trained in isolation, they can achieve a final performance accuracy comparable to that of the entire network in the same or even fewer training epochs. In particular, neurons within these winning subnets demonstrate higher rates of weight changes relative to other neurons. This observation raises the possibility of selectively reinitializing neurons that undergo minimal weight changes during training, as they contribute the least to loss function. Merely relying on the $\ell_1$ norm, which fails to capture the rate of weight changes, as described by the lottery ticket hypothesis, may not adequately capture the notion of importance. Therefore, our findings suggest that the utilization of SNIP for data-aware dynamic masking proves to be more effective, as it considers the rate of weight changes in determining the importance of parameters. This approach aligns better with the lottery ticket hypothesis and leads to improved performance and enhanced generalization capabilities in our experimental evaluations.

## 6   Conclusion

We presented Dynamic Neural Regeneration (DNR), an iterative/evolutionary training paradigm designed to improve the generalization of deep networks on small datasets. Our framework incorporates selective reinitialization at the end of each generation, employing a data-aware dynamic masking scheme to remove redundant connections based on their importance. This enables the model to increase its capacity for further learning, emphasizing the acquisition of generalizable features. Empirical results demonstrate that the proposed framework substantially enhances performance and generalization across small datasets compared to other reinitializing techniques. Moreover, DNR exhibits improved robustness in challenging real-world scenarios, including adversarial attacks and learning with class imbalances, while enhancing generalization on natural corruption data.

As a direction for future research, delving into the possibilities offered by growing networks (36) presents an intriguing path worth exploring. Our primary aim was to demonstrate the practical effectiveness of DNR, laying the foundation for theoretical exploration in this domain. We envision future studies to delve deeper into elucidating the theoretical underpinnings of DNR's success and exploring its application in diverse domains beyond those examined in this study. One possible limitation that emerged from our empirical evaluation is the sensitivity of our method to the choice of parameter selection. Addressing this limitation in future work could involve developing a robust parameter selection or importance estimation technique. This enhancement would improve our method's overall performance and deepen our understanding of overfitting dynamics by disentangling the contributions of different parameters. Finally, we believe the development of techniques like DNR holds the potential to advance the capabilities of deep learning models, paving the way for robust, adaptable, and generalizable artificial intelligence systems.

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

# A    Appendix/ Supplemental Material

## A.1    Broader Impact

We believe that the research work presented herein has the potential to significantly impact the field of deep learning by facilitating the development of models on small datasets, thus reducing the necessity for extensive data collection. This contribution is of particular significance in specific domains like autonomous navigation and medical imaging, as data acquisition can be cost-prohibitive, even in cases where labeling is unnecessary. By mitigating overfitting on small datasets, the proposed approach can improve the accuracy and reliability of deep learning models in these areas, ultimately leading to improved patient outcomes and safer autonomous systems. Moreover, this work has the potential to usher in a more efficient and sustainable use of resources in a diverse range of domains where data collection is a bottleneck.

## A.2    Evolution of Mask across Generations

In this section, we present the evolutionary process of the mask over multiple generations and its impact on the performance and generalization capabilities of the DNNs. We evaluate the effectiveness of DNR in dynamically adapting and evolving the mask throughout the training process. The ResNet18 architecture with CUB200 is used for this evaluation.

Figure 4 illustrates the evolution of the mask between generations. As training progresses, the mask undergoes iterative updates based on the data-aware dynamic masking criteria employed by DNR. The mask becomes more refined and selective with each generation, preserving important connections while pruning less relevant ones.

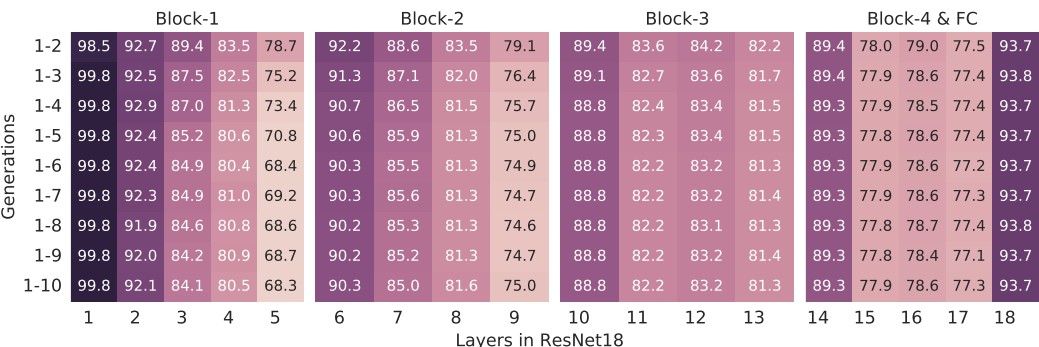

Figure 4: Layer-wise percentage overlap of the retained parameters between first and corresponding generations.

To quantify the evolution of the mask, we measure the percentage of overlap of parameters retained between the first and the corresponding generations. We observe a gradual decrease in overlap from the initial generation to subsequent generations, indicating the emergence of masks in an evolutionary training scenario. This progressive mask evolution contributes to the network's enhanced capacity for learning and generalization, evident from the test accuracy.

In conclusion, our results highlight the evolutionary nature of the mask throughout generations in the DNR framework. The dynamic adaptation and refinement of the mask lead to effective masking and improved performance and generalization of the DNN. These findings support the effectiveness of our approach in leveraging the evolutionary training paradigm to enhance the learning and generalization capabilities of deep neural networks compared to KE.

## A.3    Robustness to Natural Corruptions

In practical applications, deep neural networks often operate in dynamic environments characterized by variations such as lighting and weather conditions. Consequently, it is crucial to assess the robustness of DNNs to data distributions that undergo natural corruption. We investigate the robustness

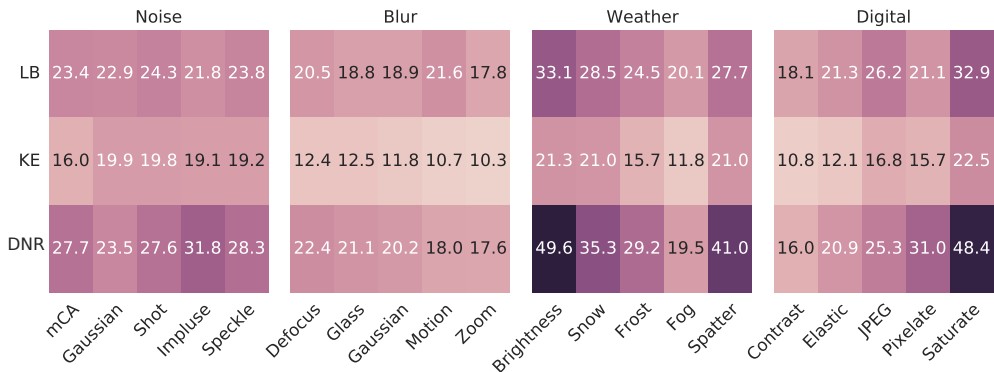

Figure 5: Robustness to natural corruptions on CIFAR10-C (37). DNR is more robust against the majority of corruptions compared to the baselines.

of DNNs to 15 common types of corruptions using the CIFAR-10-C dataset (37). Our models are trained on clean images of the CUB dataset and evaluated on CIFAR-10-C (37). To quantify the performance under natural corruption, we use the Mean Corruption Accuracy (mCA) metric.

$$\text{mCA} = \frac{1}{N_c \times N_s} \sum_{c=1}^{N_c} \sum_{s=1}^{N_s} A_{c,s} \tag{7}$$

where $N_c$ and $N_s$ represent the number of corruptions (in this case, 19) and the number of severity levels (in this case, 5), respectively. Figure 5 illustrates the average accuracy of the models across 19 different corruptions at five severity levels. In particular, our proposed method (DNR) achieves a higher mCE (27.7%) compared to the longer baseline (23.4%) and KE (16%), demonstrating its effectiveness in improving the robustness to various types of corruption. These findings highlight the benefits of selectively reinitializing network parameters using a data-aware masking approach, resulting in enhanced robustness to natural corruptions.

## A.4   Computational Cost

In our experiments, we consistently maintained a fixed training duration of 200 epochs for each generation, with the number of generations set at 10 to ensure a fair comparison. The computational cost of evolutionary training methods, including KE, LLF and DNR, scales linearly with the number of generations (T). For instance, if KE is trained for 5 generations, the total computational cost becomes 5T times that of training a single generation. To ensure a fair comparison, we trained a long baseline for the same number of epochs. The additional computational cost incurred by DNR for computing data-aware dynamic masking with SNIP is minimal. For example, on the CUB dataset with a 20% subset, this process takes approximately 20.3 seconds per generation. This calculation is performed once at the end of each generation and can be further optimized by using just 128 samples to estimate the importance without compromising performance. Appendix Table 7 demonstrates that DNR's performance is minimally sensitive to changes in the subset size.

## A.5   Robustness to Adversarial Attacks

DNNs are vulnerable to adversarial attacks, where imperceptible perturbations are added to the input during inference to deceive the network and induce false predictions (38). Therefore, we investigate the robustness of DNNs trained against adversarial attacks using the PGD-10 attack (39) on models trained on the CIFAR10 dataset. We vary the intensity of the PGD attack and evaluate the models' performance. As shown in Figure 6, our proposed framework (DNR) exhibits greater robustness to adversarial attacks at different attack strengths compared to KE and the long baseline. This highlights the efficacy of our framework in training models that can learn high-level abstractions robust to small perturbations in the input data.

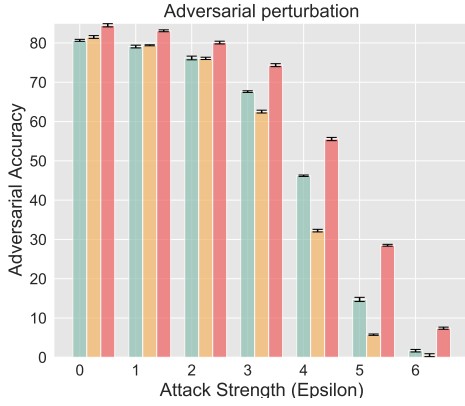

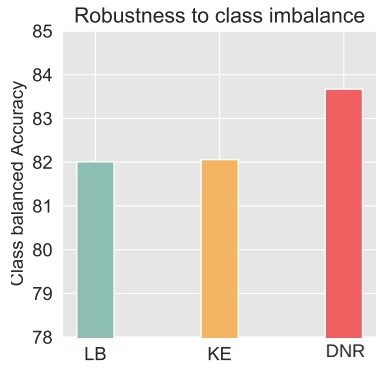

Figure 7: Robustness to class imbalance

Figure 6: Robustness to adversarial attacks

## A.6 Robustness to Class Imbalance Dataset

In real-world applications, class imbalance is a common characteristic of the input distribution, where certain classes are more prevalent than others.

This inherent class imbalance can affect the training of DNNs, as they tend to be biased towards the majority classes, thereby neglecting the minority classes (40). To address this issue, we explore the contribution of reinitialization to model training with class imbalance. We incorporate class imbalance using the power-law model on CIFAR10. The number of training samples for a class $c$ is determined by $n_c = a/(b + c)^\gamma$, where $\gamma$ represents the imbalance ratio, and $a$ and $b$ are offset parameters specifying the largest and smallest class sizes. We set a fixed gamma value of 1 in our experiments to maintain a power law class distribution. The offset parameters $a$ and $b$ are chosen such that the maximum and minimum class counts are 5000 and 250, respectively. We used balanced accuracy as a metric to measure the robustness of the model under the class imbalance scenario. Our findings in Figure 7 demonstrate that the DNR framework consistently outperforms KE and longer baselines in scenarios with class imbalance. This highlights the effectiveness of DNR in addressing the challenges posed by imbalanced class distributions and underscores its potential for practical applications.

Table 5: Performance evaluation with varying the percentage of reinitialized parameters during training using ResNet18. Test accuracy at the end of 10 generations is reported on Aircraft and CUB datasets.

| Reinit. Params (%) | Aircraft | CUB |
|---|---|---|
| 5 | 65.34 | 69.95 |
| 10 | 66.10 | 70.15 |
| 20 | **66.63** | **71.37** |
| 30 | 64.13 | 68.42 |
| 40 | 62.79 | 66.87 |

## A.7 Effect of Adjusting Reinitialized Parameter Ratios

Table 5 shows the effect of varying the number of reinitialized parameters on the performance and generalization of the model. We train the model in evolutionary settings using the DNR framework by varying different percentages of reinitialized parameters (5%, 10%, 20%, 30%, and 40%). The experiments were carried out with ResNet18. The results show that the reinitialization of a 5% percentage of parameters has no impact on performance, while the reinitialization of more than 30% has less impact on test accuracy. We find that reinitialization 20% of the parameters results in the best performance.

Table 6: Evaluating the effectiveness of the sparse model.

| Method | Full model | Sparse model |
|---|---|---|
| KE ($f_{10}$) | 66.51 | 66.21 |
| DNR ($f_{10}$) | **71.37** | **70.08** |

## A.8 Evaluating the Effectiveness of the Sparse Model

In this section, we assess the effectiveness of the sparse model (containing 20% fewer parameters than the full/dense model) obtained through the selective neurogenesis process during the inference phase. We examine the sparse and dense models' test performance compared to the original KE framework. For this, we measure the performance of the ResNet18 model trained on CUB200. Table 6 presents the accuracy results obtained by the sparse model compared to the dense model. Surprisingly, despite the considerable reduction in the number of parameters, the sparse model achieves comparable accuracy compared to the dense model in the DNR framework. Furthermore, DNR demonstrates superior performance in both the full and sparse model scenarios compared to the KE. This indicates that the selective neurogenesis process successfully retains the critical connections necessary for accurate predictions while eliminating redundant or less informative connections. Our evaluation demonstrates that the sparse model obtained through the selective neurogenesis process offers several benefits during inference. It maintains high accuracy while achieving improved computational efficiency compared to the KE. These results highlight the practicality and efficacy of leveraging selective neurogenesis to create efficient and compact deep learning models that can be readily deployed in real-world scenarios.

## A.9 Varying the quantity of data used for Importance estimation

In our experiments, we randomly sampled 20% of the dataset to estimate the importance of the parameters after the end of each generation. Here, we analyze the impact of the number of data used to determine the important estimation on the final performance. Similar to Lee et al. (23), we used as few as 128 samples to estimate the important parameters using SNIP. Table 7 shows that DNR is not sensitive to the variation in the input data used to estimate the importance as the final performance remains unchanged.

Table 7: Evaluation with varying the quantity of data for importance estimation. Test accuracy at the end of 10 generations is shown on Aircraft and CUB datasets.

| | # samples | Aircraft | CUB |
|---|---|---|---|
| DNR | $0.2\,|\mathcal{D}|$ | **66.63** | **71.37** |
| | 128 | 66.45 | 71.26 |

## A.10 Experiments on Medical dataset

To explore the performance of deep learning in low-data medical settings, we conducted experiments using ResNet-50 on the Papila dataset(41). This dataset, with its limited size and complex features, is representative of the challenges in medical imaging. The Papila dataset is a medical imaging dataset focused on analyzing and identifying ocular structures, particularly the optic disc and retinal layers. We conducted this experiment using ResNet-50 with a learning rate of 0.001, batch size of 32, and trained for 3 generations (1 generation = 200 epochs). For DNR 20% of the parameter is reinitialized using SNIP. Our preliminary results suggest that Dynamic Network Reconfiguration (DNR) shows promise in the medical domain when compared to baseline training. However, further hyperparameter tuning is necessary to fully optimize performance.

While initial findings are encouraging, we highlight the need for more detailed studies in reinitialization-based methods within the medical domain on different datasets. Future work should address these challenges to improve generalizability across low-data medical imaging datasets.

Table 8: Performance of ResNet-50 on Papila Dataset: Comparison of DNR and Baseline in terms of Sensitivity, Specificity, and Accuracy.

|  | Sensitivity | Specificity | Accuracy |
|---|---|---|---|
| Baseline ($f_3$) | 56.11 | 58.68 | 56.90 |
| DNR ($f_3$) | 57.03 | 57.83 | 58.20 |

Table 9: Performance of DNR across generations on the Flower dataset

| **Generations** | **Accuracy (%)** |
|---|---|
| 10 | 68.36 |
| 20 | 72.10 |
| 30 | 73.56 |

## A.11 Performance of DNR Across Generations on Flower Dataset

As the number of training generations increases, the performance of DNR begins to saturate. This indicates diminishing returns in accuracy improvement with extended training. This saturation effect is corroborated by our analysis of the mask evolution across generations, as illustrated in Figure 2. The overlap percentage of the mask progressively increases, with a higher overlap observed between the 9th and 10th generations compared to the 1st and 2nd generations. This trend suggests that the mask becomes more saturated and stable, aligning with the model's convergence to a lower-loss landscape.

## A.12 Summary of Datasets and Implementation Details

Taha et al. (5) employs various image resizing techniques for different datasets; however, they do not provide specific details about the resizing parameters in their paper. To ensure consistency across our experiments, we resize all datasets to a fixed size of (256, 256). Moreover, to fine-tune the hyperparameters, we utilize a validation split, and the reported results are based on the test set whenever it is available.

Table 10: Details of the five used classification datasets.

| Datasets | Classes | Train | Validation | Test | Total |
|---|---|---|---|---|---|
| CUB-200 (26) | 200 | 5994 | N/A | 5794 | 11788 |
| Flower-102 (25) | 102 | 1020 | 1020 | 6149 | 8189 |
| MIT67 (27) | 67 | 5360 | N/A | 1340 | 6700 |
| Aircraft (29) | 100 | 3334 | 3333 | 3333 | 10000 |
| Standford-Dogs (28) | 120 | 12000 | N/A | 8580 | 20580 |

For experiments on large datasets, we used the following settings. The experiments were conducted on three different datasets: CIFAR-10/100, Tiny-ImageNet. For CIFAR-10/100, the training was performed for 160 epochs. A batch size of 64 was used, along with a step-based learning rate scheduler. The learning rate decay was applied between epochs 80 and 120, with a decay factor of 10. The momentum was set to 0.9, and l2 regularization was applied with a coefficient of 5e-4. The initial learning rate used was 0.1. There were no warmup epochs in this case.

For the Tiny-ImageNet dataset, the training was also conducted for 160 epochs. The batch size was reduced to 32, and a step-based learning rate scheduler was used. Similar to CIFAR-10/100, the learning rate decay occurred between epochs 80 and 120, with a decay factor of 10. The momentum and l2 regularization were set to 0.9 and 5e-4, respectively. Additionally, 20 warmup epochs were applied. Throughout all experiments, a resetting ratio of 20% is used for all generations. All the training and evaluation are done on the NVIDIA RTX-2080 Ti GPU. The time required to approximately train 10 generations of DNR on CUB200 with ResNet18 is approximately 1.68 hours. It is worth mentioning that to compare our method with other baselines, we utilized the results

presented in the KE paper (5) as a point of reference. For the hyperparameters used in training small datasets, please refer to Section 4.

