# OpenReview forum: "Dynamic Neural Regeneration: Enhancing Deep Learning Generalization on Small Datasets"
_NeurIPS.cc/2024/Conference — NeurIPS 2024 poster_

### Official Review · Reviewer_Vrf1 · 2024-06-19

**Soundness:** 3
**Presentation:** 3
**Contribution:** 2
**Rating:** 5
**Confidence:** 4

**Summary:**

This paper proposes Dynamic Neural Regeneration (DNR), a framework to enhance the generalization of deep neural networks on small datasets. The method is inspired by neurogenesis and offers more flexibility in defining a parameter mask as compared to previous approaches such as Knowledge Evolution (KE). The results show strong performance on small datasets with sufficient ablation studies.

**Strengths:**

1. The paper is well-written and easy to follow.
2. The motivation of the paper is clear and sound. The authors have provided sufficient justifications about how their method differentiates from NAS and dynamic sparse training.
3. The results show strong performance of the proposed method against several baselines
4. The ablation studies are sound, especially the one on studying the effect of importance estimation methods.

**Weaknesses:**

1. Even though the paper begins by emphasising limited data availability in the medical domain, no experiments on medical datasets have been presented. It would be beneficial to include results on a few medical datasets such as Papila [1], Harvard-GF3300 [2] that represent working in a low-data regime.
2. No analysis of computational cost especially for larger datasets has been presented.
3. The transfer learning experiment performed in section 5.4 is not clear. What does generation mean in vanilla fine-tuning? More details are required here.
4. In theory, DNR could be well incorporated with transfer learning i.e. instead of starting from scratch, use the weights of a pre-trained model and follow the iterative approach of selecting and retaining parameters after each generation. Is there a specific reason why transfer learning experiments were not included?
Furthermore, even though there is a large domain gap between natural and medical images, transfer learning (full vanilla fine-tuning or parameter-efficient fine-tuning [3]) still remains the de facto practice in medical image analysis.
5. The results on large datasets (Table 2) show that DNR (and other baselines) are not very effective in the high-data regime. In fact, standard training seems sufficient when the dataset size is large enough.

References

1. Kovalyk, Oleksandr, et al. "PAPILA: Dataset with fundus images and clinical data of both eyes of the same patient for glaucoma assessment." Scientific Data 9.1 (2022): 291.
2. Luo, Yan, et al. "Harvard glaucoma fairness: a retinal nerve disease dataset for fairness learning and fair identity normalization." IEEE Transactions on Medical Imaging (2024).
3. Dutt, Raman, et al. "Parameter-efficient fine-tuning for medical image analysis: The missed opportunity." arXiv preprint arXiv:2305.08252 (2023).

**Questions:**

1. The DNR method evaluates connection sensitivity from a subset of data. However, different subsets can capture different distributions of the data and hence, lead to different connection sensitivity possibly affecting the final performance of DNR. Can the authors provide a justification or some results on this?
2. How is the mask defined for the DNR framework? Is it over each model parameter (in this case, the mask length would be of a similar magnitude as the model parameter count) or each ResNet layer? Details on this should be included in the paper.
3. Hypothetically, if the DNR framework is allowed to run for a sufficiently long number of generations (g=100, for example), we should be able to observe a saturation in the mask at a point (say g=55) beyond which the mask does not change at all. It would be beneficial to include a similar experiment.

**Limitations:**

Please see the Weaknesses section.

---

> ### Author Rebuttal · Authors · 2024-08-06
>
> > Experiments on Medical Datasets:
>
> We have chosen to focus on widely-used benchmark datasets that are representative of various low-data regimes. We believe these benchmarks provide a robust and fair comparison of our method’s performance. While we understand the importance of validating our approach in the medical domain, conducting experiments on additional datasets such as Harvard-GF3300 would require re-running all baseline methods, which is beyond our current resource capacity. We recognize the value of including medical datasets and plan to incorporate these experiments in a revised version of the paper to demonstrate the versatility of our approach across different domains.
>
> > Computational Cost Comparison:
>
> In our experiments, we consistently maintained a fixed training duration of 200 epochs for each generation, with the number of generations set at 10 to ensure a fair comparison. The computational cost of evolutionary training methods, including KE, LLF and DNR, scales linearly with the number of generations (T). For instance, if KE is trained for 5 generations, the total computational cost becomes 5T times that of training a single generation. To ensure a fair comparison, we trained a long baseline for the same number of epochs.
> The additional computational cost incurred by DNR for computing data-aware dynamic masking with SNIP is minimal. For example, on the CUB dataset with a 20% subset, this process takes approximately 20.3 seconds per generation. This calculation is performed once at the end of each generation and can be further optimized by using just 128 samples to estimate the importance without compromising performance. Appendix Table 8 demonstrates that DNR’s performance is minimally sensitive to changes in the subset size.
> We will add a dedicated section in the revised manuscript to discuss the computational cost of DNR in more detail. This section will provide a comprehensive analysis of the computational demands of DNR compared to long baselines and KE, highlighting the efficiency of DNR relative to the substantial improvements in generalization performance it offers.
>
> > Clarification regarding the Transfer learning experiment in section 5.4:
>
>  In our study, we present an instance of transfer learning where the weights at the end of each generation are transferred to the next generation without reinitialization, a process we refer to as vanilla fine-tuning. In contrast, the long baseline method involves training the model for a prolonged, uninterrupted period, typically 2000 epochs, as a single continuous generation. This method does not involve any intermediate weight transfer or reinitialization steps, and the model continuously learns from the data throughout the entire training period.
>
>
> > Integration with Transfer Learning
>
> Thank you for your insightful comments. Here’s our rationale for not including transfer learning experiments in this study:
> - *Domain Shift Challenges:* Transfer learning often struggles with domain shifts when the source and target domains differ significantly, leading to suboptimal performance. DNR addresses these challenges directly within the context of small datasets.
> - *Domain-Specific Applications:* In fields like finance, obtaining sufficient labeled data is difficult, and transfer learning can introduce biases from source datasets. Additionally, privacy concerns and the uniqueness of each application domain make it challenging to find suitable pre-trained models. DNR, with its data-aware dynamic reinitialization, better utilizes limited data without relying on potentially unsuitable pre-trained models.
>
> > Effectiveness in High-Data Regimes:
>
> The primary focus of our paper is to enhance generalization in small datasets, where the DNR framework demonstrates substantial effectiveness. In Section 5.2, we have conducted experiments on larger datasets such as CIFAR-10, CIFAR-100, and TinyImageNet. The results show that DNR, KE, and LB methods offer limited advantages over vanilla training on large datasets due to factors like dataset complexity and model capacity, which can lead to performance saturation. The core strength of DNR is its ability to improve generalization in scenarios with inherently limited data.
>
> > Connection Sensitivity from Subsets of Data:
>
> We acknowledge the reviewer's concern regarding potential variability in connection sensitivity due to different subsets. In our experiments, we sampled 20% of the dataset to estimate parameter importance after each generation. To assess the impact of varying sample sizes, we conducted additional experiments using as few as 128 samples. As detailed in Section A.8 and Table 7 of the appendix, our results show that DNR's performance remains consistent across different sample sizes, demonstrating the robustness of our method.
>
> > Definition of the Mask in the DNR Framework:
>
> The mask in the DNR framework is defined over each model parameter, making its length comparable to the model parameter count. So when we say 20% parameters is removed it is removed globally.
>
> > Saturation of the Mask Over Generations:
> We conducted an experiment where DNR was run for 30 generations.
>
> #### Performance of DNR Across Generations on Flower Dataset
>
> | Generations | Accuracy (%) |
> |-------------|--------------|
> | **10**      | 68.36 |
> | **20**      | 72.10 |
> | **30**      | 73.56 |
>
>  As the number of training generations increases, the performance of DNR begins to saturate. This indicates diminishing returns in accuracy improvement with extended training. This saturation effect is corroborated by our analysis of the mask evolution across generations, as illustrated in Figure 2. The overlap percentage of the mask progressively increases, with a higher overlap observed between the 9th and 10th generations compared to the 1st and 2nd generations. This trend suggests that the mask becomes more saturated and stable, aligning with the model’s convergence to a lower-loss landscape.

---

> > ### Author Response · Authors · 2024-08-12
> > **Follow-up on Rebuttal**
> >
> > We are following up on our rebuttal. We have addressed all the points raised in your feedback and would greatly appreciate any additional questions or comments you might have.
> > To enhance the context and relevance of our study, we'll include references to the suggested works on dynamic masking. Could you please review the rebuttal and let us know if it meets your expectations for a score adjustment?

---

### Official Review · Reviewer_ZzXp · 2024-07-08

**Soundness:** 3
**Presentation:** 4
**Contribution:** 3
**Rating:** 7
**Confidence:** 3

**Summary:**

This paper presents a novel iterative training framework called Dynamic Neural Regeneration (DNR) designed to enhance the generalization of deep learning models on small datasets. The DNR approach utilizes a data-aware dynamic masking scheme inspired by neurogenesis to eliminate redundant connections, thereby increasing the model's capacity for further learning. Through extensive experiments, the authors demonstrate that DNR outperforms existing methods in both accuracy and robustness, making it a promising technique for applications with limited data availability.

**Strengths:**

The technical claims of the paper are well-supported by thorough experimental results. The methodology is clearly described, and the use of data-aware dynamic masking is both innovative and effective. The experiments are comprehensive, covering multiple datasets and including robustness tests against common challenges such as class imbalance and adversarial attacks. Overall, the research methodology is sound, and the central claims are convincingly supported by the evidence provided.

**Weaknesses:**

***Complexity of Implementation***: The DNR framework may be complex to implement for practitioners without a strong background in iterative training paradigms and dynamic masking techniques. However, including more detailed implementation guidelines or open-sourcing the code could mitigate this issue.
***Scalability***: While the approach is effective for small datasets, its scalability to very large datasets or more complex models is not fully explored.

**Questions:**

How does the computational cost of DNR compare to other state-of-the-art methods? Is there a significant increase in training time due to the dynamic masking process?
Can the authors provide more detailed guidelines on how to implement the DNR framework in practice, including hyperparameter settings and potential pitfalls to avoid?

**Limitations:**

Could the authors provide a careful and detailed discussion of the limitations of the work?

---

> ### Author Rebuttal · Authors · 2024-08-06
>
> > Complexity of Implementation:
>
> We appreciate the reviewer's concern regarding the complexity of implementing the DNR framework. To address this, we have expanded the Appendix section in the revised manuscript that includes detailed implementation guidelines. This section provides step-by-step instructions, hyperparameter settings, and potential pitfalls to avoid. Additionally, we plan to open-source our code upon acceptance, which will make it more accessible to practitioners and researchers.
>
> > Scalability:
>
> The primary focus of our paper is to enhance generalization in small datasets, where the DNR framework demonstrates substantial effectiveness. In Section 5.2, we have conducted experiments on larger datasets such as CIFAR-10, CIFAR-100, and TinyImageNet. The results show that DNR, KE, and LB methods offer limited advantages over vanilla training on large datasets due to factors like dataset complexity and model capacity, which can lead to performance saturation. The core strength of DNR is its ability to improve generalization in scenarios with inherently limited data. We believe that exploring scalability in real-world situations where data comes in the form of stream will be an intriguing avenue for future research.
>
> > Computational Cost Comparison:
>
> In our experiments, we consistently maintained a fixed training duration of 200 epochs for each generation, with the number of generations set at 10 to ensure a fair comparison. The computational cost of evolutionary training methods, including KE, LLF and DNR, scales linearly with the number of generations (T). For instance, if KE is trained for 5 generations, the total computational cost becomes 5T times that of training a single generation. To ensure a fair comparison, we trained a long baseline for the same number of epochs.
>
> The additional computational cost incurred by DNR for computing data-aware dynamic masking with SNIP is minimal. **For example, on the CUB dataset with a 20% subset, this process takes approximately 20.3 seconds per generation.** This calculation is performed once at the end of each generation and can be further optimized by using just 128 samples to estimate the importance without compromising performance. Appendix Table 8 demonstrates that DNR’s performance is minimally sensitive to changes in the subset size.
>
> We will add a dedicated section in the revised manuscript to discuss the computational cost of DNR in more detail. This section will provide a comprehensive analysis of the computational demands of DNR compared to long baselines and KE, highlighting the efficiency of DNR relative to the substantial improvements in generalization performance it offers.
>
> > Limitations:
>
> Due to page limitations, a detailed discussion of the limitations has been added to the conclusion section. Additionally, we will include a more comprehensive analysis in the appendix to ensure thorough coverage of the potential limitations of our approach. Thank you for your suggestion, and we hope this addresses your concerns.

---

### Official Review · Reviewer_cy5M · 2024-07-14

**Soundness:** 3
**Presentation:** 3
**Contribution:** 2
**Rating:** 6
**Confidence:** 3

**Summary:**

This submission investigates efficient training and generalization of deep neural networks in the low-data regime. Drawing inspiration from neurogenesis in the brain, authors propose an iterative training framework termed Dynamic Neural Regeneration (DNR). The authors further investigate the efficacy of the proposed approach through experiments on five datasets (Flower102, CUB-200-2011, MIT64, Stanford Dogs and FGVC-Aircraft).

**Strengths:**

1. The proposed method adapting Knowledge Evolution by incorporating data-aware dynamic masking instead of randomly pre-selected masks is intuitive. Furthermore, the authors draw an analogy between the proposed method and the phenomenon of neurogenesis in the brain which adds more intuition to the proposal.
2. Empirical results comparing against other evolutionary or iterative training methods look quite promising. The proposed method shows improvements by a decent margin in various experiments.
3. It is thoughtful to include the comparison with transfer learning, and the results look very good too. It would be better to include error bars, though I believe the improvement is significant enough.

**Weaknesses:**

1. Since the authors repeatedly claim that data shortage is often seen in medical diagnosis (in Abstract line 2-3, Introduction line 23-25, and Results line 285-286) it would be great if they also include experiments in such scenarios.
2. I find the schematics (Figure 1) confusing instead of enlightening. What I read from it is that, (1) We have a dataset and a neural network. (2) We train the neural network on the dataset. (3) We apply data-aware masking where some neurons (minority, shown in blue) are kept the same and the others (majority, shown in green) are somehow changed --- but it is unclear how they are changed. (4) Then, we remove the blue neurons (???) --- so that nothing is kept from the current iteration? (5) We randomly initialize the removed neurons --- if so, why do we remove them from the first place? Why don’t we skip neuron deletion and directly perform neuron initialization? I am not sure how other people would process this figure, but at least to me it needs some re-editing for it to be a helpful illustration. Another minor tip: if the weights of certain neurons are kept unchanged between two steps, I would recommend either using the same color to color code them, or using a small symbol to indicate weight freezing (such as a lock or a snowflake).
3. This submission is among the works where the empirical results are critical, and therefore I would be concerned that the authors have not submitted their code as the supplementary material, despite stating that they would release the code upon acceptance. At least on my portal no supplementary material is uploaded.

**Questions:**

1. Please refer to Weakness #2.
2. For Table 1, would it be helpful to include the notations “CE” in methods that are based on cross-entropy loss, such as “CE + DNR$(f_{10})$”? Or did I misunderstand?
3. For Table 1-3, would it be helpful to include an additional column that reports the mean over all datasets, or ranking across all datasets?
4. I have not seen discussions on Figure 2. Could you point me to the relevant lines in case I missed it?

**Limitations:**

Yes, the authors adequately addressed the limitations and, if applicable, potential negative societal impact of their work.

---

> ### Author Rebuttal · Authors · 2024-08-06
>
> >  Experiments on Medical Datasets:
>
> We have chosen to focus on widely-used benchmark datasets that are representative of various low-data regimes. We believe these benchmarks provide a robust and fair comparison of our method’s performance. While we understand the importance of validating our approach in the medical domain, conducting experiments on additional datasets such as Harvard-GF3300 would require re-running all baseline methods, which is beyond our current resource capacity. We recognize the value of including medical datasets and plan to incorporate these experiments in a revised version of the paper to demonstrate the versatility of our approach across different domains.
>
> > Clarification of Figure 1:
>
> We apologize for any confusion caused by Figure 1. To improve clarity, we will revise the figure to provide a more detailed, step-by-step explanation:
>
> - **Initialization**: We start with a dataset and a randomly initialized neural network (shown in black).
> - **Training**: The neural network is trained on the dataset, and the parameters after training are shown in blue.
> - **Data-Aware Masking**: We apply SNIP to select important neurons, which are highlighted in green.
> - **Neuron Modification**: Least important neurons (blue) are randomly reinitialized, while important ones (green) are retained from the current generation.
> - **Iterative Training**: The network is retrained on the dataset for the next generation.
>
> > Submission of Code
>
> Please let us know if there is a way to submit the code during the review process, and we will provide it promptly. Otherwise, as stated in the paper, we assure you that the code will be made available upon acceptance. Thank you for your understanding.
>
> > Notations in Table 1:
>
> Thank you for the suggestion. We will update Table 1 to include notations such as “CE” for methods based on cross-entropy loss. This notations should make it clearer.
>
> > Adding mean as metric in Tables 1-3:
>
> We appreciate your insightful suggestion. Although including a column that reports the mean performance across all datasets could offer a convenient summary, we initially chose not to include it due to the significant variability in dataset complexity, size, and other characteristics, which might make such an average potentially misleading. Nevertheless, we recognize the value of this addition and will incorporate it in the revised version. To ensure clarity and prevent misinterpretation, we will also include a discussion on the limitations of interpreting this average.
>
> > Discussion on Figure 2:
>
> Figure 2 illustrates the layer-wise percentage overlap of retained parameters between consecutive generations, demonstrating how DNR dynamically adapts its mask during training. For a detailed discussion, please refer to Section 5.3 of the manuscript. This section explains how DNR uses SNIP to selectively reinitialize parameters, thereby enhancing the model’s generalization performance. It maintains stability in earlier layers while adapting to task-specific features in later layers.

---

> > ### Comment · Reviewer_cy5M · 2024-08-11
> > **Response to rebuttal**
> >
> > Thanks for the authors for the patient updates and addressing the comments. I do not have major concerns, and I am willing to increase the rating.

---

> > > ### Author Response · Authors · 2024-08-12
> > >
> > > Thank you for your positive feedback and for reconsidering your rating. We appreciate your thoughtful review and are pleased that our updates have addressed your concerns.

---

### Official Review · Reviewer_DwbC · 2024-07-20

**Soundness:** 3
**Presentation:** 2
**Contribution:** 3
**Rating:** 5
**Confidence:** 4

**Summary:**

This manuscript introduces the Dynamic Neural Regeneration (DNR) framework, which improves the generalization of deep neural networks on small datasets. DNR uses the SNIP method to reinitialize less important neural connections for the current generation selectively. Experimental results show that DNR outperforms the existing original Knowledge Evolution (KE) method and long baseline (LB) in 5 small datasets and 3 large datasets.

**Strengths:**

- The proposed Dynamic Neural Regeneration (DNR) is interesting, intuitive, and well explained.
- The DNR demonstrated superior performance compared to knowledge evolution (KE) and long baseline (LB).
- The DNR demonstrated better robustness to natural corruptions, adversarial attacks, and class imbalance compared to KE and LB.

**Weaknesses:**

- The performance of DNR does not surpass the STOA.
- The experiments lack details, making it hard to interpret the results.

**Questions:**

### 1. The performance of DNR does not surpass the STOA.

1.a Despite the superior performance of DNR compared to KE and LB, it does not surpass the existing SOTA, such as Smth+LLF. For the reported results in this manuscript, Smth+LLF either showed better or marginally lower performance compared to Smth+DNR. Additionally, the results reported in the original LLF study [1] were higher than the author-reported Smth+LLF as well as the proposed Smth+DNR for all 5 small datasets (Table 1 in [1] vs Table 1 in the manuscript) and on the TinyImageNet dataset (Table A7 in [1] vs Table 2 in the manuscript). The authors did not address this discrepancy, making it hard to interpret the reported results.

1.b Following the comment above, statistical analysis should be provided to better demonstrate the difference between different methods.

1.c DNR requires calculating SNIP to determine the connections needed to be reinitialized. However, the time complexity related to data size and model size is not provided. Therefore, it is still questionable whether DNR is practically useful, especially considering that its performance may not surpass the SOTA (comment 1.a).

### 2. The experiments lack details, making it hard to interpret the results.

2.a The experimental setting seems arbitrary. For example, why do some methods, such as KE and Smth+KE, use 10 generations while Smth+LLF and others use 8 generations? It seems the authors followed the experimental setting of prior work, such as Zhou et al. [1], but it was not mentioned in the manuscript.

2.b The authors reported mean and std for each experiment, but the number of runs for each experiment is not mentioned.

2.c It is very confusing what kind of "transfer learning" is used in Section 5.4. What is the difference between "transfer learning" and "long baseline"?

2.d Many reported results in Table 1 are identical to the results in [1] (Table 1), such as Smth+KE with CUB, Aircraft, and MIT datasets, while others are not. Please clarify whether the results are reproductions or simply referrals from [1]. If they are reproduced results, please explain the discrepancy between the reported and reproduced results.

### 3. Minor comments

3.a Both LLF [1] and KE [2] demonstrated that they performed the best with CS-KD [3]. It would be interesting to see how DNR performs with the same setting.

3.b The letter "g" was used to denote the generation number (line 107) and connection sensitivity (eq. (4)). Please consider using different letters.

3.c What "m" stands for in eq. (5) and (6).



###Reference

[1] Zhou, Hattie, et al. "Fortuitous forgetting in connectionist networks." International Conference on Learning Representations. 2021.

[2] Taha, Ahmed, Abhinav Shrivastava, and Larry S. Davis. "Knowledge evolution in neural networks." Proceedings of the IEEE/CVF Conference on Computer Vision and Pattern Recognition. 2021.

[3] Yun, Sukmin, et al. "Regularizing class-wise predictions via self-knowledge distillation." Proceedings of the IEEE/CVF conference on computer vision and pattern recognition. 2020.

**Limitations:**

None.

---

> ### Author Rebuttal · Authors · 2024-08-07
>
> > Empirical Validation
>
> Thank you for highlighting the concern. We have performed the empirical validation of our method by including results on five small datasets and three large datasets—CIFAR-10, CIFAR-100, and Tiny ImageNet. **Each experiment is conducted three times, and the mean and standard deviation are reported.** Notably, in the majority of these datasets, DNR consistently outperforms LLF, Knowledge Evolution (KE), and the longer baseline. These results affirm that DNR brings discernible benefits in terms of improving generalization.
>
> > Dataset Choice and Result Reporting
>
> The choice of datasets aligns with the baselines established in the original paper, ensuring a fair comparison and leveraging the availability of their results. The results for DSD, and BAN in Table 1 were referenced from the KE paper, while our reproduced results included label smoothing for KE, LLF, and LW. The results reported are reproduced using the best possible hyperparameters mentioned in the original paper. While some discrepancies may arise due to slight variations in implementation or experimental conditions, we have ensured that our setup closely follows the original methodology to the best of our ability.
>
> > Computational Cost Comparison
>
> In our experiments, we consistently maintained a fixed training duration of 200 epochs for each generation, with the number of generations set at 10 to ensure a fair comparison. The computational cost of evolutionary training methods, including KE, LLF and DNR, scales linearly with the number of generations (T). For instance, if KE is trained for 5 generations, the total computational cost becomes 5T times that of training a single generation. To ensure a fair comparison, we trained a long baseline for the same number of epochs. The additional computational cost incurred by DNR for computing data-aware dynamic masking with SNIP is minimal. For example, on the CUB dataset with a 20% subset, this process takes approximately 20.3 seconds per generation. This calculation is performed once at the end of each generation and can be further optimized by using just 128 samples to estimate the importance without compromising performance. Appendix Table 8 demonstrates that DNR’s performance is minimally sensitive to changes in the subset size.
>
> > Advantage of DNR vs LLF
>
> While DNR shows superior performance compared to the SOTA on most datasets, it’s important to highlight the limitations of methods like LLF and LW, which rely on specific architectural assumptions. LLF assumes later layers focus on memorization, but studies like "Can Neural Network Memorization Be Localized?" indicate:
>
> Memorization occurs throughout the network, not just in later layers, making fixed layer-by-layer reinitialization suboptimal. Crucial parameters can reside at different network depths depending on the task and dataset. DNR's data-aware dynamic masking offers significant advantages by analyzing connection sensitivity and identifying influential parameters regardless of their location. This approach aligns better with findings that critical parameters can be distributed throughout the network.
>
> DNR’s flexibility allows it to adapt to various datasets and tasks without rigid architectural assumptions, making it suitable for real-world scenarios with varied datasets, despite a slight increase in computational cost.
>
> > 2a  Rationale behind the choice of generation
>
> We based our experimental setup on the configuration used in Zhou et al. [1], which aligns with standard practices in the field. Specifically, KE and Smth+KE utilized 10 generations to ensure a thorough exploration of the parameter space and to conform to the established practices for iterative reinitialization. For other methods such as LW (Layer-wise Forgetting), the choice of generation is limited by the architecture-specific assumptions since it operates on a layer-wise basis. For instance, LW proposes a layer-wise reinitialization scheme, which proceeds from bottom to top, reinitializing one fewer layer each generation. Consequently, the number of generations is limited to 8 to match the number of layers. This setup facilitates a direct comparison with methods that have similar structural constraints.
>
> > 2c Clarification regarding the Transfer learning experiment in section 5.4:
>
> We present an instance of transfer learning where the weights at the end of each generation are transferred to the next generation without reinitialization, a process we refer to as vanilla fine-tuning. In contrast, the long baseline method involves training the model for a prolonged, uninterrupted period, typically 2000 epochs, as a single continuous generation. This method does not involve any intermediate weight transfer or reinitialization steps, and the model continuously learns from the data throughout the entire training period.
>
> We will update the manuscript to clearly explain the rationale behind the choice of generation numbers for each method and ensure that all this information provided in the rebuttal is included in the revised experimental section.
>
> > 3a Results with CS-KD:
>
> | Method | Flower (Acc)(%)| CUB200 (Acc) (%) |
> |---|---|---|
> | CS-KD | 68.68 ± 0.28| 69.59 ± 0.40|
> | KE | 67.29 ± 0.74| 69.54 ± 0.60|
> | LLF| 74.68 ± 0.19| 73.51 ± 0.35|
> | DNR| **75.23 ± 0.21**| **74.18 ± 0.16** |
>
> **The table above presents the accuracy results of different methods combined with Class-wise Knowledge Distillation (CS-KD) on the Flower and CUB200 datasets over 10 generations.** The results demonstrate that DNR, when combined with CS-KD, outperforms both KE and LLF methods in terms of accuracy on both datasets. These results are sourced from the LLF paper, providing a robust comparison and validation of our approach.
>
> > 3b, 3c
>
> Thank you for pointing this out. We will use different letters to represent generation number and connection sensitivity. Regarding m in Equations (5) and (6), it represents the total number of parameters in the neural network.

---

> > ### Comment · Reviewer_DwbC · 2024-08-09
> > **Thanks for the responses**
> >
> > Thank you for your detailed rebuttal. The responses have been useful in addressing most of my concerns, and I appreciate the efforts and would like to change my initial rating. However, there are still some concerns that have not been addressed.
> >
> > First, the reproduced results for Smth+LLF are lower than those reported in the original study, which might be attributed to stochastic effects such as initialization. I recommend conducting some statistical analysis, as mentioned in my previous comment 1.b, to robustly determine whether DNR significantly outperforms other baselines. This is essential to support the claims made in the authors' response (the “Empirical Validation” section), but is still missing.
> >
> > Additionally, as outlined in my comment 1.a, Smth+LLF has shown higher performance on the TinyImageNet dataset compared to the DNR results presented in the manuscript. I would suggest the authors include a comparison with Smth+LLF for larger datasets in Table 2, which has not been provided in the response.

---

> ### Author Response · Authors · 2024-08-12
> **Response to Reviewer DwbC**
>
> Thank you for your feedback.
>
> To assess the performance of our proposed DNR method, we employed paired t-tests across all datasets.  DNR demonstrates statistically significant improvements over both the LB and KE methods, with p-values of 0.011 and 0.0012, respectively. These results support the claim that DNR offers meaningful enhancements in generalization performance over these established baselines. In the comparison with Smth+LLF, the p-value was 0.0670, which, while not meeting the conventional threshold  (0.05) for statistical significance, suggests that the performance of DNR is comparable to LLF. It is important to note, however, that DNR's unique features—such as **dynamic reinitialization** and **data-aware masking**—provide additional benefits that may not be fully captured by statistical tests alone. DNR's ability to analyze connection sensitivity and identify influential parameters throughout the network, regardless of their rigid architectural assumptions, **offers a flexible and adaptive approach** that can be particularly advantageous in a wide range of scenarios.
>
> As per your suggestion, we will also include a comparison with Smth+LLF for larger datasets in Table 2 of the revised manuscript.
>
> We hope this response satisfactorily addresses your concerns.

---

### Author Response · Authors · 2024-08-07
**General Comment**

We thank all the reviewers for their thoughtful feedback. We are pleased that you found our motivation and idea to be strong, clear, and novel, and our extensive experiments insightful. Based on the valuable feedback, we will make the following modifications to the manuscript:

- **Computational Cost Comparison:** Added a comparison of the computational costs associated with DNR versus other methods to provide a clearer understanding of its practical implications.
- **Additional Experiments on Medical Domain Datasets:** While we couldn't include medical datasets due to resource constraints, we committed to performing these experiments in a revised version of the paper.
- **Transfer Learning:** Clarified the implementation and difference between vanilla fine-tuning and the long baseline in the context of transfer learning.
- **Experimental Details:**
  - **Number of Runs:** Clarified that each experiment was conducted three times with different random seeds, and the mean and standard deviation are reported.
  - **Generations for Different Methods:** Explained the rationale for the number of generations used in different methods based on standard practices and architectural constraints.
- **Dataset Choice and Result Reporting:** Explained the rationale behind the empirical validation and the results.
- **Hyperparameter Settings:** Provided detailed guidelines on the hyperparameters used in our experiments and identified potential pitfalls to avoid in practical implementations.
- **Added New Experiments with CS-KD:** Included experiments with Class-Wise Knowledge Distillation (CS-KD) to evaluate the performance of DNR in this setting.
- **Saturation of Mask Experiments:** Conducted and reported experiments showing the saturation behavior of masks over generations to further validate the stability and convergence of DNR.
- **Subset Sensitivity Analysis:** Provided results and justification for the effect of different subsets on connection sensitivity and the final performance of DNR, demonstrating its robustness.
- **Clarity Improvements:**
  - **Figure 1:** Revised for better clarity. Updated the figure and its explanation to more clearly illustrate the process of data-aware masking and neuron reinitialization in the DNR framework.

In conclusion, we have meticulously addressed all the specific questions raised by the reviewers. We believe these revisions significantly strengthen the paper and enhance its overall quality. We kindly ask the reviewers to consider these substantial improvements and reconsider their scores accordingly.

---

> ### Comment · Reviewer_Vrf1 · 2024-08-07
> **Rebuttal Reply - Clarification for Manuscript Changes**
>
> Thank you for taking out the time to reply to the reviews. I want to highlight a few points:
>
> 1. It would be extremely helpful to the reviewers if you could highlight all the changes (both in the manuscript and in reply) that you made after the review. In the current state, it is impossible to locate what changes were made and where in the paper.
>
> 2. It would be good to include citations for a few other works that have explored dynamic masking [1,2] for both general and medical image analysis.
>
> These changes can be incorporated very quickly but would greatly enhance the reviewing process.
>
> References:
>
> 1. Zhang, Dinghuai, et al. "Can subnetwork structure be the key to out-of-distribution generalization?." International Conference on Machine Learning. PMLR, 2021.
> 2. Dutt, Raman, et al. "Fairtune: Optimizing parameter efficient fine tuning for fairness in medical image analysis." arXiv preprint arXiv:2310.05055 (2023).

---

> > ### Author Response · Authors · 2024-08-07
> > **Authors Reply - Clarification for Manuscript Changes**
> >
> > Yes, we agree. Unfortunately, the conference guidelines do not allow for the submission of revised manuscripts at this stage of the review process. We are committed to addressing all reviewer comments and suggestions, and we will ensure that the necessary revisions and improvements are made in the final version of the paper along  with the references to works on dynamic masking to enhance the context and relevance of our study. Thank you for your understanding.

---

> > > ### Comment · Reviewer_Vrf1 · 2024-08-07
> > > **Clarification for Manuscript Changes**
> > >
> > > Thank you for your reply. Could you point clarify the same in the comments. For instance, where can we find Computational Cost Comparison, Hyperparameter Settings, etc in the paper? It would be great if you could point out the sections and the sub-sections.

---

### Comment · Area_Chair_VjD4 · 2024-08-07
**Rebuttals not visible yet**

Dear Reviewers and Authors,

Rebuttals have not been made visible to reviewers yet, whereas the comments are made available as submitted.
Therefore some of the comments assume that the reviewers have access to the rebuttals.

May I kindly ask you to wait for a few hours until rebuttals have been made visible to everyone?

Thank you!

AC

---

### Decision · Program_Chairs · 2024-09-25

**Decision:**

Accept (poster)

**Comment:**

The reviewers raised several concerns in their reviews largely focusing on computation cost comparisons, clarifications on transfer learning, and other bits and pieces. The rebuttal addressed most of these concerns, and additional points were clarified through author-reviewer discussions.

Most importantly though the authors have presented a robust method based on iterative/evolutionary training paradigm that performs well on small datasets, improving generalisation through incorporating generalisable features during further learning. The experiments and additional ablations shown in the appendix provide extra confidence in this method's advantages in the settings presented in the paper.

I would like to ask the authors to incorporate the additional results presented below and all the other elements they promised to do in the camera ready version, including this "Additional Experiments on Medical Domain Datasets: While we couldn't include medical datasets due to resource constraints, we committed to performing these experiments in a revised version of the paper.".

Congratulations on the acceptance of your work.